# Measurement of Energy Access Using Fuzzy Logic

**Diego Seuret-Jimenez [1], Tiare Robles-Bonilla [1],*  and Karla G. Cedano [2]**

[1] Centro de Investigación en Ingeniería y Ciencias Aplicadas, Universidad Autónoma del Estado de Morelos, Ave. Universidad 1001, Cuernavaca 62209, Morelos, Mexico; dseuret@uaem.mx

[2] Instituto de Energías Renovables, UNAM, Xochicalco s/n, Azteca, Temixco 62588, Morelos, Mexico; kcedano@ier.unam.mx

* Correspondence: tiare.roblesbno@uaem.edu.mx

**Abstract:** This paper describes an innovative method to evaluate energy access in any of size population by applying fuzzy logic. The obtained results allow ranking regions of Mexico according to their overall energy access. The regions were determined by the country's political division (32 states). The results presented herein are in close correspondence with other studies undertaken. This method is recommended because it is possible to use as an assessment tool due to its representativeness—that is, it poses a heuristic alternative to quantify the level of Energy Access in a particular region through qualitative data. It is also efficient and cost-effective in terms of computer resources. This is extremely important to public policy makers that require more accurate, faster and cheaper methodologies to assess energy access as an indicator of well-being.

**Keywords:** energy access; energy use; fuzzy logic

## 1. Introduction

Understanding the way in which people use energy at home is necessary to move forward in the development of public policies which foster more efficient energy usage. Several surveys have been developed to measure energy access and its use. Butera et al. [1] developed a study about Brazil (Rio De Janeiro), in which two cities were analysed on energy access and the level of energy poverty through questionnaires carried out in 400 households. This helped to determine the local living conditions and the availability of basic energy services, as well as explore the actual energy access and energy poverty in the favelas. One of its main findings was that electricity consumption is very high compared to the service provided—as much as Italian or German households, which are much richer—in addition to electricity access being threatened by interruptions and low tension. This method is replicable with small adaptations; however, Butera et al. do not use fuzzy logic. Jimenez et al. [2] performed an analysis of surveys to determine the state of the electrification barriers in Latin America. Taking three variables—household income, household location, and the country's level of economic development—they analyse 12 countries in Latin America (Bolivia, Brazil, Chile, Costa Rica, Dominican Republic, Ecuador, Guatemala, Honduras, Mexico, Peru, Paraguay and El Salvador). The study shows serious inequality in electricity access, a family living in a poor country has a lesser chance of accessing electricity than a family with the same income but living in a richer country. This study does not use fuzzy logic, but it shows the application of a mathematical method—regression analysis.

In Mexico, for example, there is the Household Expenditure and Income Survey, measuring [3], among other things, energy services and expense in Mexican households. Another, of recent implementation, is the National Survey on Energy Consumption in Private Households (ENCEVI, its Spanish acronym) [4]. This was designed to help better understand the existing relationship between people and energy. Nevertheless, these exercises of information gathering do not provide simple and

reliable tools for researchers and policy makers to compare and understand energy access and use at a household level. It is necessary to run the data sets produced by these surveys through often costly processing systems that require large and precise data sets to model energy access.

Addressing and measuring energy access is a complex issue. In most indicator sets that are used to measure energy poverty or that are related to basic energy services, the closest indicator is absolute electricity access. For instance, the World Economic Forum reports a 100% with respect to electrification rate in 2018 for Mexico [5]. It also includes another relevant indicator, electricity supply quality, measured by the following question: "In your country, how reliable is the electricity supply (lack of interruptions and lack of voltage fluctuations)?" on a scale from 1 to 7, 1 being "extremely unreliable", 7 being "extremely reliable", for which Mexico scored 4.9. This might be interpreted as "reliable". This indicator was subsequently measured differently in the following edition, also published in 2018 [6], as the percentage of electricity losses (comprising transmission, distribution and non-technical losses). Even though the indicator has the exact same name, it reflects very different things. However, the World Economic Forum's double definition of electricity access shows that the former version of the indicator was far more representative as the impact that the quality of the service had (or the perception from the consumer). This is the main reason behind our decision to measure Energy Access (EA) by asking a question about the availability of energy services to the population.

It is important to highlight that we are not addressing energy poverty (or fuel poverty) in this paper. While both energy poverty and energy access are related, energy poverty goes beyond the availability (or the perception of the availability) of any given energy service, but also energy use and the social behaviours that accompany said use. This is clearly defined by Thomson et al. [7], who state that energy poverty occurs "when a household is incapable of securing a degree of domestic energy services (such as space heating, cooling, cooking) that would allow them to fully participate in the customs and activities that define membership in society".

As part of your desk research, we did a small scientometric analysis using the Web of Science database. Using the phrase "energy access" as search criteria, we found 778 articles with it in either the title, key words or abstract. Simultaneously, we looked for articles with the phrase "fuzzy logic" and found 48,227. Nevertheless, when we intersected the searches, we could not find a paper that talked about both energy access and fuzzy logic. Therefore, this might present a novel methodology for looking at this issue.

Lotfi A. Zadeh [8] defined fuzzy logic in the 1960s for issues regarding language. It has now been applied more broadly to a diversity of knowledge areas, such as the control of automatic electro-mechanical processes [9], human activity control [10], decision making [11], etc. Nobre et al. [12] consider fuzzy logic a computational and mathematical framework suitable to represent approximate reasoning. It takes into consideration everyday life concepts, experiences, observations, etc., with all of them having "fuzzy limits" [13]. Tron y Margaliot [14] describe fuzzy logic as an effective methodology for creating models by considering intuition and agent related behaviour.

Fuzzy logic has been used in economic topics related to energy. Spandagos et al. [15] state that in order to understand the factors that foster consumer energy behaviour and thus enable the development of more efficient polity, it is necessary to create energy consumption models that take economic behaviour into consideration. With this in mind, Spandagos developed a model based on fuzzy logic that includes concepts of bounded rationality, time discounting of gains and pro environmental behaviour. The model is developed from the decision perspective, rules based on human reasoning and behaviour, and also takes into consideration currency, personal comfort and environmental responsibility related variables to generate predictions regarding purchasing decisions and air conditioning use. An important similarity was found between the results generated by the model and historic data on energy usage for the cooling of urban populations. This proved it to be a trustable model. Spandagos showed the feasibility of using fuzzy logic to combine economic, physical quantitative data with qualitative concepts.

Among the several applications of fuzzy logic, there is a model to define and measure sustainability called SAFE, proposed by Phillis et al. [16]. In this model, fuzzy logic is used as well as 75 indicators to classify 128 countries, also considering expert opinions, international agreements and frameworks. This model measures sustainability on a world scale, but it can be adapted to smaller regions since its variables, both input and output, rules and membership functions can be modified. Fuzzy logic has also been used in the evaluation of energy systems in dwellings, as was done by Gamalath et al. [17]. In their paper they propose an assessment framework for the condition of the energy system in multi-unit residential buildings (MURB). Their evaluation method applies fuzzy logic to overcome data uncertainty and imprecision. It also uses the rules to combine different performance categories to obtain a grade on the general condition of a MURB. The application of fuzzy logic can help to account for qualitative data that might be obtained in stakeholders' consultations. Their study demonstrated that fuzzy logic can be used to improve the strategies of asset management and operation of existing buildings.

In a similar manner to Spandagos et al., that achieve a model based on fuzzy logic with concepts both quantitative as qualitative, our work combines qualitative values with energy expenses. Our model is also adaptive, as like that of Phillis et al., because the variables (input and output), the rules and the membership functions can be modified in the light of each country's context. Furthermore, like the Gamalath et al. model, ours can be used to design public policy as well as to improve management strategies.

Fuzzy logic can obtain results from human language, as opposed to other methods, if the variables and their values are presented as quantifiable data. This mathematical tool is helpful for highly complex systems that cannot be represented by differential equations or which cannot be solved through conventional means, as their solution entails a high level of complexity. However, Fuzzy Logic does not require complex mathematical models, but anyone with expertise on a given subject can use the methodology, i.e., it is a heuristic tool. Within the bivalent logic of Charles Boole [18], an element from the whole might be part or not of the whole, using Zadeh logic being a part of a fuzzy whole is neither one nor zero, but gradually varies between one and zero.

When trying to solve a system using fuzzy techniques, there are three things the expert needs to define. Firstly, the properties which will characterize the system (linguistic variables), and the set of values which they will undertake (linguistic values). Having established these, they must define the membership functions linking these properties with the system. The next and last key step is what makes fuzzy logic a great approach for evaluating complex systems, which is the rule definition. These are defined by an expert performing the analyses and include their biases on how the system should behave. To quantify the membership on a given group does not represent either a probability or a percentage, but rather how a given characteristic places us in a group we are referring to a population group. This allows us to implement a frequently used concept in Fuzzy Logic, the membership in a group with specific characteristics. Defining linguistic variables and rules is what brings about a system that can be adjusted from different perspectives. Every fuzzy analysis is unique, as each expert will attach their personal imprint.

For this study, we used data from ENCEVI, the survey conducted by National Institute of Statistic and Geography (INEGI, its Spanish acronym) [4], which gathers data provided by the persons interviewed. It is important to mention that there are no direct measurements involved in this survey. As defined by the fuzzy logic methodology, we needed to select the linguistic variables for the system. These were selected for a population by its energy access. We chose transport, cooking fuel and electricity expenditure, as they all had an associated measure and are of key importance to an individual's well-being. According to Sovacool et al. [19] " ... for both the rural and urban poor, low mobility—regard less of the technology or mode of transport involved—stifles the attainment of better living standards. It reduces the ability to earn income, strains economic resources, and limits access to education and health services and markets ... ". It is with this consideration that we include transport as a variable related to energy access. Both cooking fuel and electricity expenditure have been used

on previous studies done on energy access [20] and Energy Poverty [21] in Mexico. Furthermore, electricity access at this time of world development is crucial, as it provides numerous benefits in addition to other services being closely related, such as entertainment, education, communication, etc. [22]. Furthermore, as we know, the massive trend towards electrification will make it more and more relevant to individual and community wellbeing. Another important reason for choosing these variables was considering that all of them can be measured in the same unit (Mexican Peso).

## 2. Theoretical Basis

When doing a fuzzy logic analysis, the first step is to define linguistic variables, which will be the criteria framing the system. For our energy access analysis, we have defined them as: transport, cooking fuel and electricity expenditure. They serve as indicators to evaluate or characterize it, and are made up of values which we call linguistic values. Linguistics values are then subdivided into bands. For the case presented herein, we have set them as follows: low, middle, and high.

The second step is the rule design. For this, we must first calculate the number of rules to be applied in assessing a problem. This is calculated by the expression $A=B^C$. In which A: number of rules. B: number of linguistic variables. C: amount of bands.

Our energy access index will be comprised of 3 linguistic variables, framed within 3 value bands, requiring 27 rules. The number of rules is the first filter to understand if fizzy logic is the right method to solve the problem. It also needs to take into consideration the degree of knowledge the expert might have about the problem, and their capacity to come up with sensible rules. Having a high number of rules will increase the processing time. If we had chosen to evaluate 5 linguistic variables—each one with 5 bands—the system would require 3125 rules ($5^5$).

The membership function is a very important element in problem solution. It shows the degree to which an element is related or has a characteristic associated to a linguistic value. It defines membership. This function might be of different types, fundamentally it could be triangular, trapezoidal, Z type and S type—we have chosen to use the latter for this analysis [3].

*2.1. Description of Experimental Data.*

To have homogenized linguistic variables, the defined set of linguistic values can all be expressed in monetary units. Data at state level were obtained from the ENCEVI survey performed in 2018 by INEGI [4]. Figures 1–3 show the averages by state for transport, fuel used in cooking and electricity expenses.

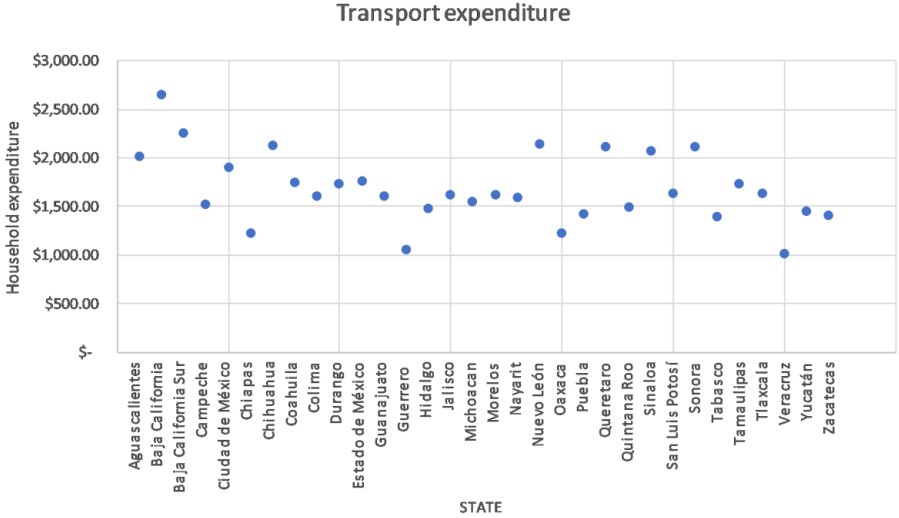

**Figure 1.** Average monthly transport expenditure per household in pesos, by State. Source ENCEVI, INEGI.

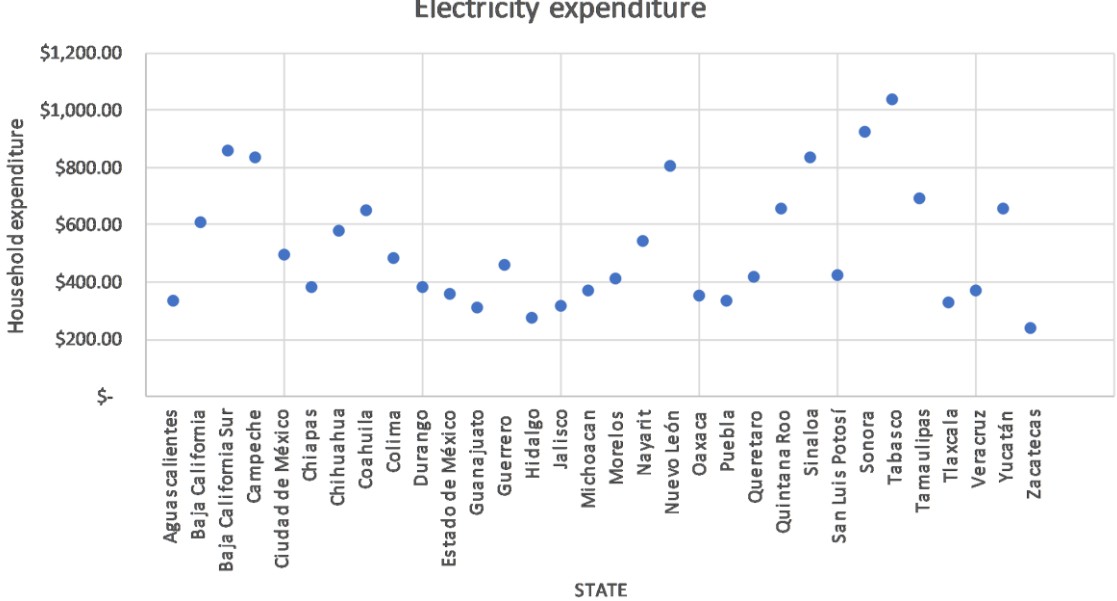

**Figure 2.** Average bimonthly electricity expenditure per household in pesos, by State. In Mexico, payment for domestic electric service is made every two months. Source ENCEVI, INEGI.

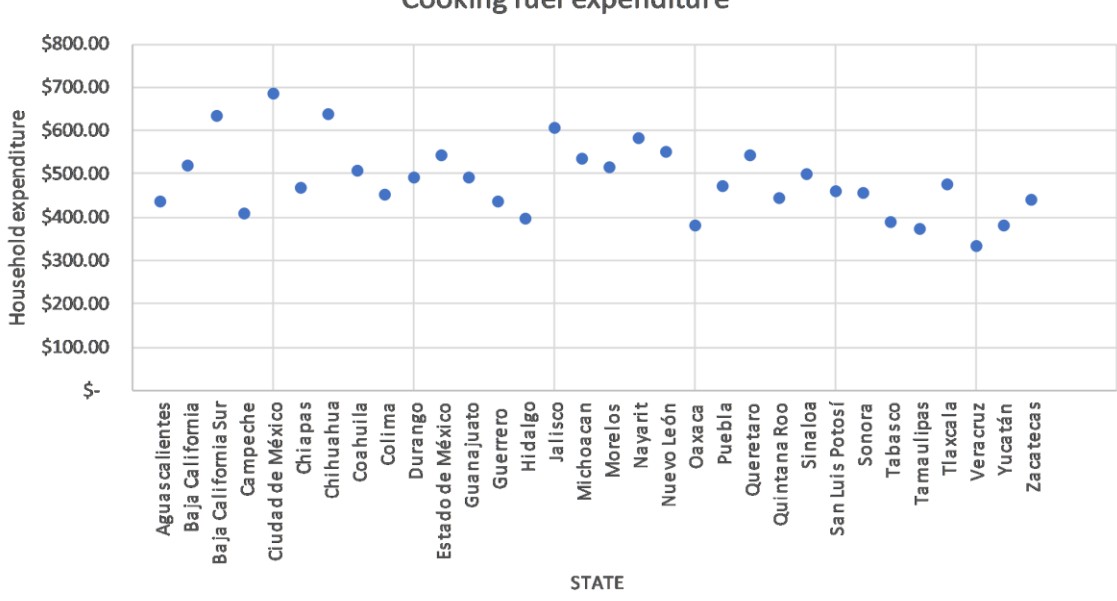

**Figure 3.** Average monthly cooking fuel expenditure per household in pesos, by State. Source ENCEVI, INEGI.

It is important to mention that these variables, taken from the literature, are coupled to the variables of the survey. Within the survey the expense for household appliances or heating is not included, this is included in the total expenditure of electricity.

In this case, the ENCEVI survey does not consider electricity as a cooking fuel. In Mexico, very few households use electric stoves. The selected linguistic variables show similar behaviour in all states. This could be a problem for the precision of the method. The value-bands need to be adjusted to refine the results (Energy Access) value-range. This procedure will also depend on the experience of the expert in charge of designing the process.

## 2.2. Calculus

Based on each state's population sample average expenditure, we define the range for the categories each value can fall into. The values can fall into three categories—high, medium and low—and a number is given to each one. For the present case study, the input variables are shown in Figures 4–6 and the Figure 7 represent the output variable.

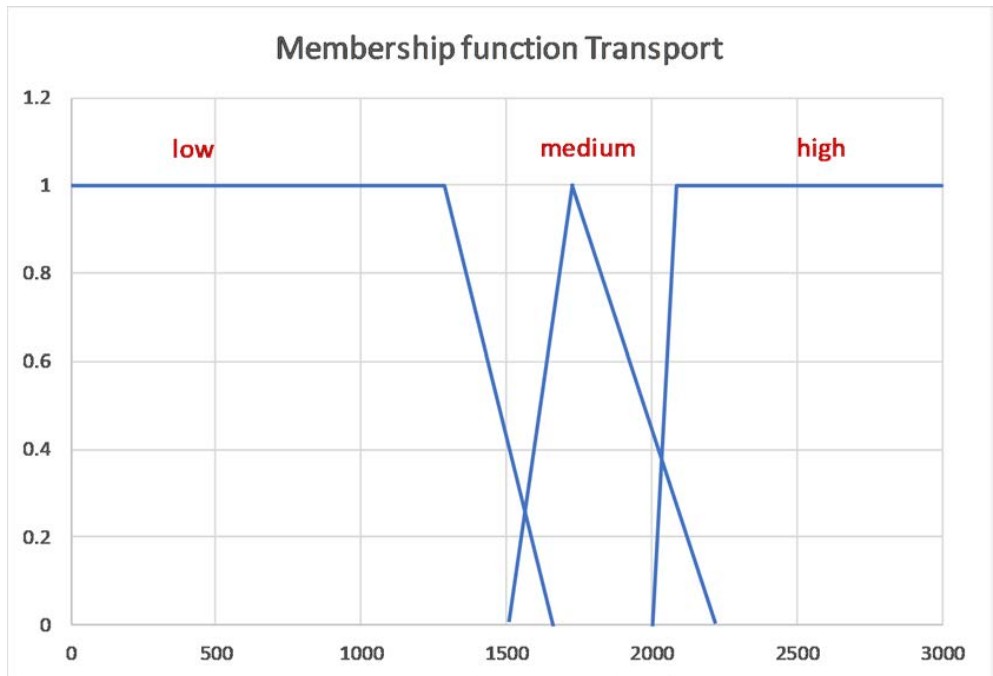

**Figure 4.** Membership function transport.

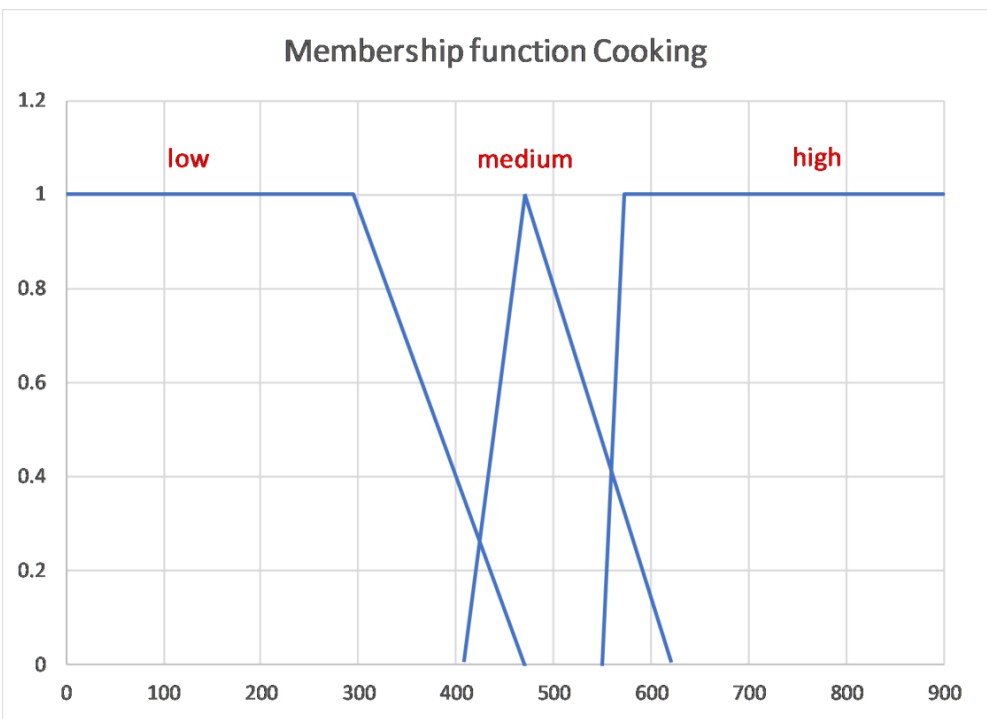

**Figure 5.** Membership function cooking.

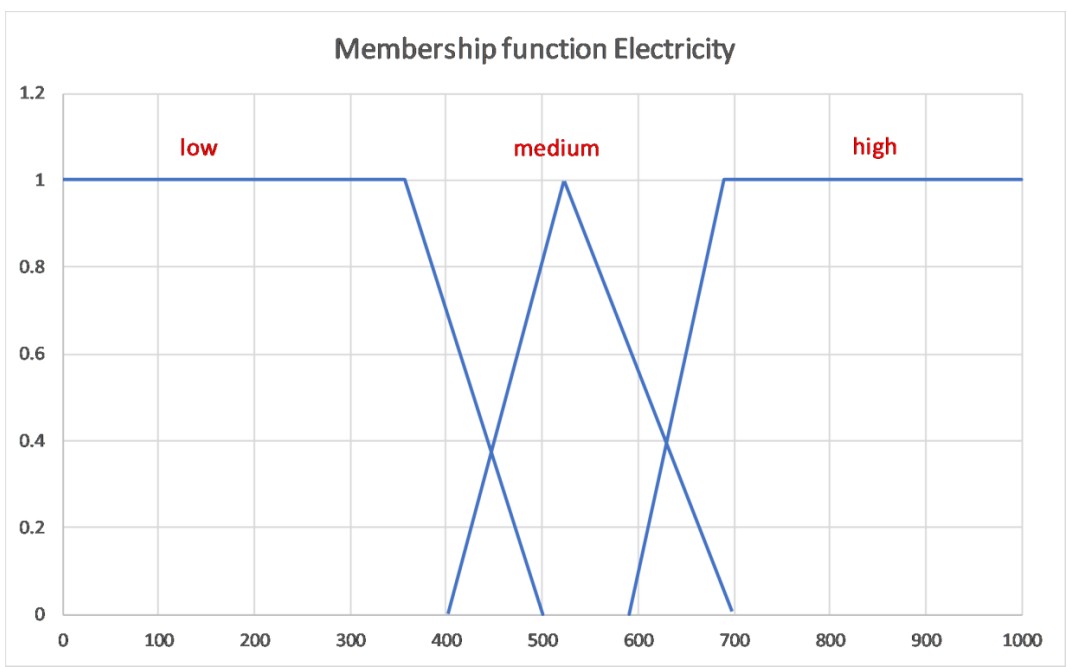

**Figure 6.** Membership function electricity.

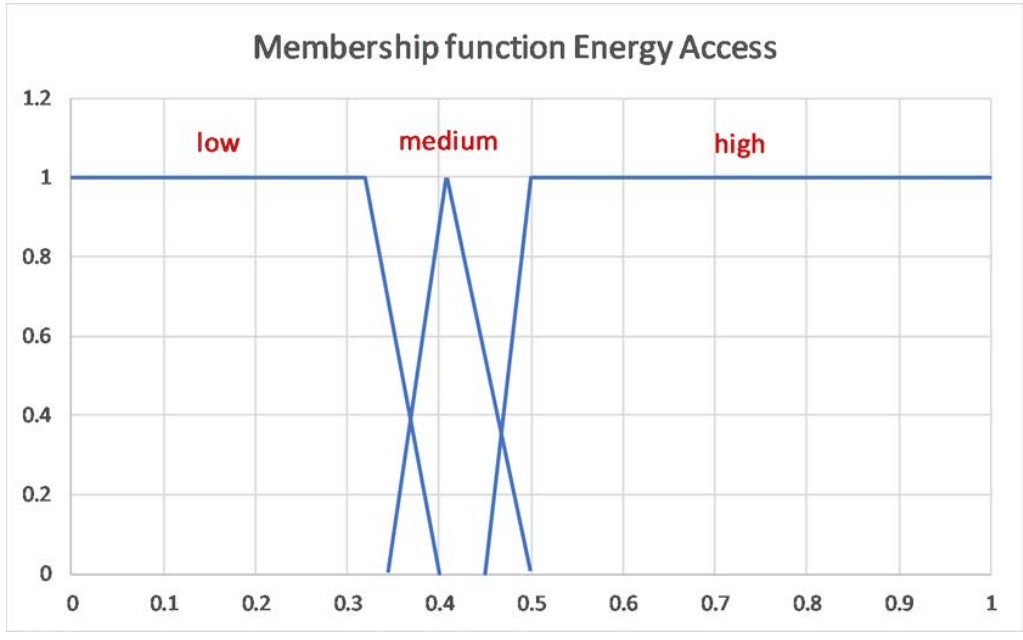

**Figure 7.** Membership function EA (Energy Access).

In the Figure 7 the values closer to 0 indicate a low energy access, while values closer to 1 indicate a high availability to energy services.

The definition of the values for the membership function "Energy Access" were obtained using a similar method to Nussbaumer et al. [22] to measure energy poverty. They used a threshold to define energy poverty of 0.32, based on experience and through intensive tests. It is a well-known fact that membership function values can be modified according to the context of the specific application. We defined our values due to the similar behaviour that the results from the fuzzy logic algorithm gave when comparing the per capita PIB of each state with the reality of each state. However, those values on energy access in each country should be assigned by experts so to have a model that truthfully represents any given regional context. The experts on the topic are those with the knowledge, the

expertise and the access to high quality data, that can determine whether or not a model represents each countries energy landscape.

The next step after defining linguistic variables, values and membership functions is to draft the rules, then to undertake an analysis of each element. This procedure can be performed manually. However, there are several tools that can do this task in a more efficient manner. For this study, we used a tool designed in MATLAB [23].

The same procedure used in defining "EA" output values is used, this output has three variables: high, medium and low, with 0.46, 0.41 and 0.32 values respectively. Below the presentation of a typical rule is shown.

If (transportation is high) and (cooking is low) and (electricity is medium) then (AE is medium).

Using the above rule and the other 26 rules, we designed the system by applying the Mamdani fuzzy inference systems, which closely recreates both human reasoning and the fuzzy if-and-then rules. Moreover, we used the Mamdani method as it generates a fuzzy set as its output. This more complete output is the reason we chose this over more popular methods such as the Sugeno method, whose output is only linear or constant.

It is important to mention that, just as with the linguistic values, rules can be modified depending on the person making the analysis. This presents an important advantage in comparison with other analysis methodologies, as it allows us to assess the system under other conditions.

## 3. Results

Once the system is ready and the expenditure averages are defined, you can start calculating the value of energy access for each region. Figure 8 shows how the tool works, by inputting the values we need to be analysed and showing the energy access value as a result. For the example below, we introduced Mexico City's values for transportation, cooking and electricity (1901, 682 and 496 respectively). The tool shows us the EA value is of 0.762.

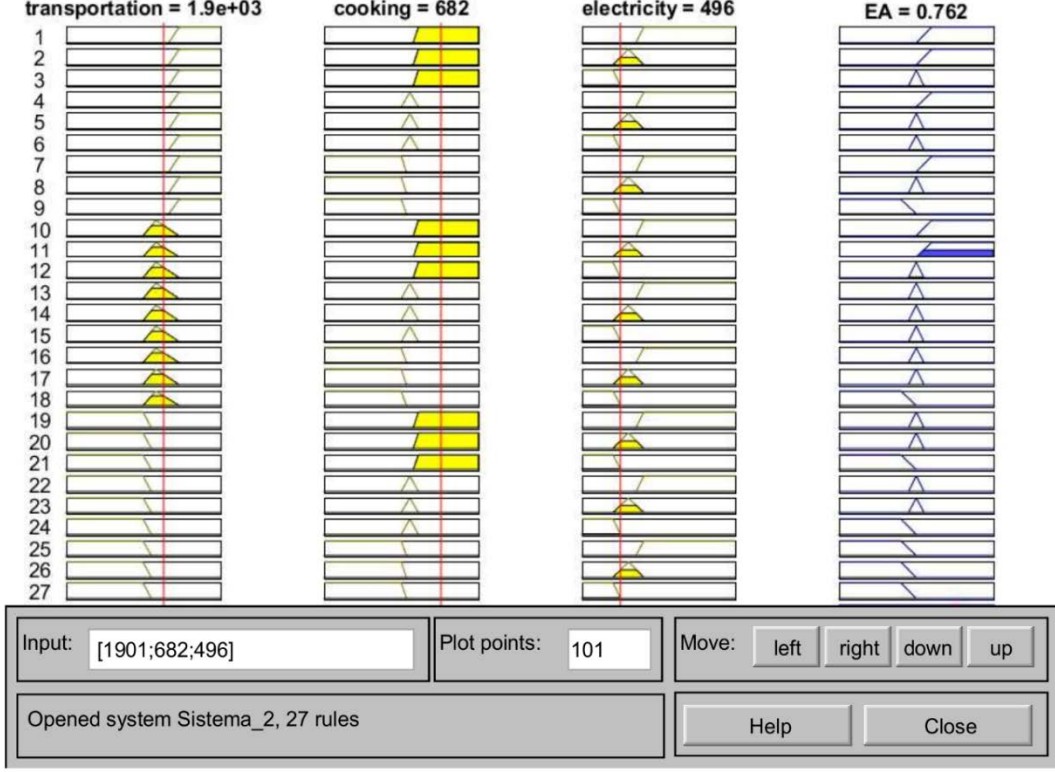

**Figure 8.** MATLAB screen results. Evaluation of the rules and their results to determine energy access.

Figure 9 shows energy access across all states; as per our previous definition, states over 0.46 are classified as having a high energy accessibility, and those under 0.32 as low accessibility, which means that in general, they will face above average difficulties in gaining access to electricity, cooking fuels and transport compared to the other states.

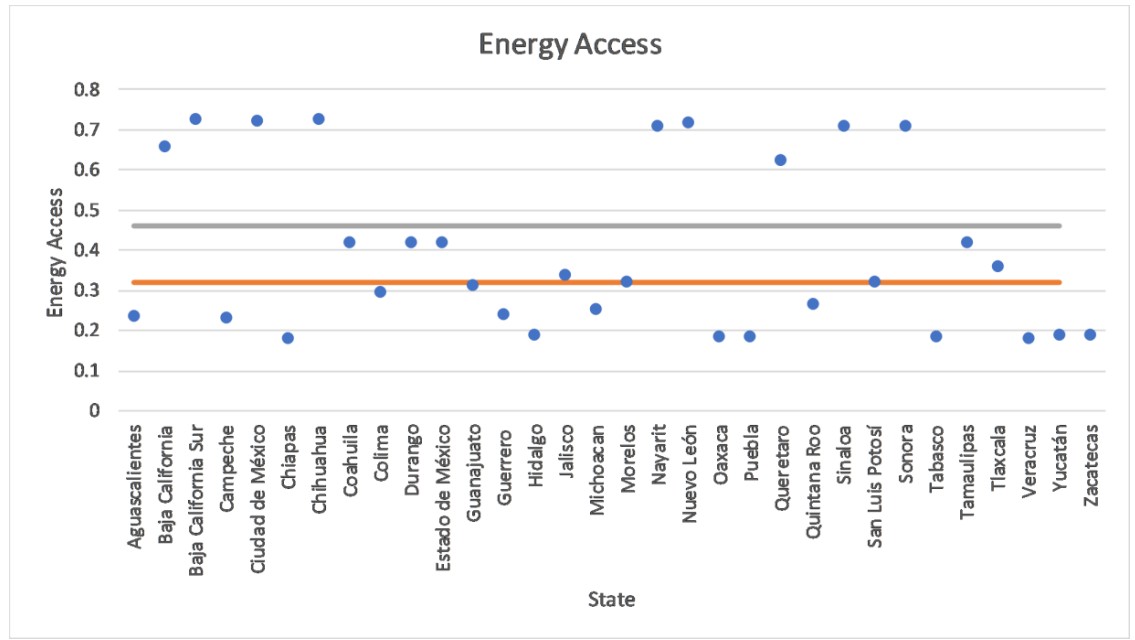

**Figure 9.** Energy access by state. Less than 0.32 is considered as low energy access, greater than 0.46 is considered as high energy access. Between 0.32 and 0.46 is considered as medium energy access.

## 4. Discussion

We obtained a distribution of energy access (EA) across Mexico by applying fuzzy logic. As it is a fuzzy set, it can be divided in different ranges or zones. This division varies in response to local information on energy access and the characteristics of the locality. Since it is a heuristic criterion, the most valuable use of the tool is to monitor Energy Access in a locality through time. When comparing the values between regions, other socioeconomic indicators are needed in order to have a better understanding of each region's relationship with energy access. This is not only a strictly theoretical absolute result; rather, it is a methodology that enables comparisons. In this specific case, when we associate a membership function to a number, this element within the whole state reflects the specific property to which we are referring—in this case EA. Each of the EA ranges might be called: Low EA, Medium EA and High EA. The same rule will be used to classify all of them.

Drawing comparisons with research from other contexts is challenging, as the study of social phenomena is complex and involves a different approach. In natural sciences, defining magnitudes and models to estimate a given situation it is a straightforward matter. However, in social sciences, although exact mathematical models are applied to social issues, there are many variables intervening. Furthermore, in most cases, all the different variables and conditions that take place are unknown. So, the application of these type of tools is helpful to grasp the context of various entities, and to start the understanding of social phenomena. Social applications have a complex nature; describing and modelling them is a challenge requiring a complex system approach. For our case study, we wanted to see the relationships between EA and three factors—economic development, geographic characteristics and socio-cultural behaviour. Figure 10 shows our comparison between EA and GDP per capita, which we calculated using 2017 data from INEGI [24] and CONAPO [25].

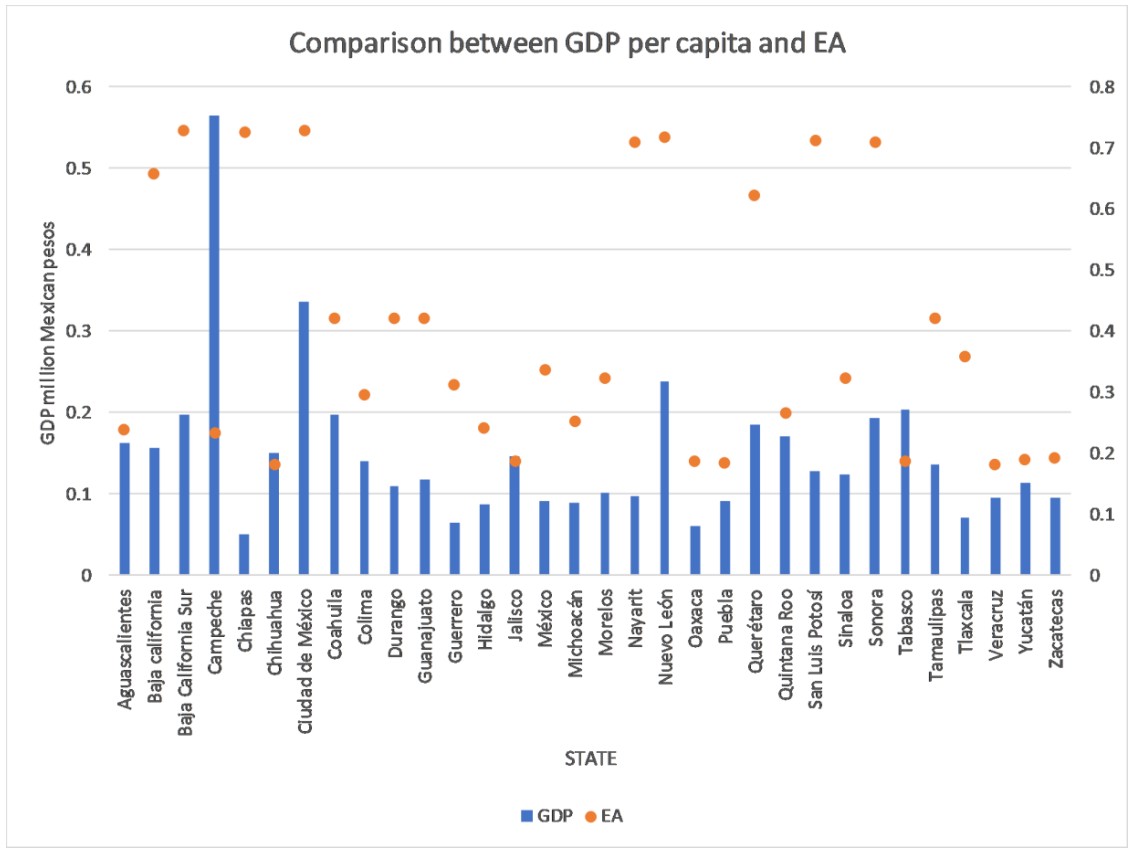

**Figure 10.** Comparison between GDP per capita and EA by state.

The analysis of this graph takes only one aspect related to energy access into consideration, the overall economic activity per state. Even though the prosperity in each region has a close relationship with the quality of energy access and the socioeconomic level of the population, it is clear that there are several factors that we need to analyse to fully understand each individual case. In general, we can see that the level of EA is relatively equal to or greater than the GDP per capita in almost all of the regions. This is because the largest and most important energy and utilities companies (Pemex and CFE) are public (owned by the state). Therefore, there is a natural bias of those public companies to promote social well-being. However, both Campeche and Tabasco pose an exception to this trend. Campeche is the "richest" state because most of the oil production is based there. However, that wealth is not part of the economic activity of most of the population. Since Pemex, the public oil company, is responsible for that income, GDP per capita does not reflect the economic behaviour of the inhabitants of Campeche. The EA indicator shows an average value, not related to GDP per capita. In Tabasco, something similar happens; it is the second most important oil production location. On the other hand, we have the case of Chiapas, an extremely poor region that has a very important indigenous population and is well known for having a strong political agenda. Its inhabitants have access not only to the services provided by the public energy companies, but they have a long-standing tradition of using biomass, so this analysis portrays the reality of Mexico's landscape regarding EA.

If we were to add geographic characteristics and socio-cultural habits, then this analysis would be even more complex. Many authors prefer to draw their analysis based solely on the "geographic regions" variable, as if the only important factor might be the geography of the place, and they do not take the socioeconomic reality into consideration. As we have already expressed, the solution to social problems includes many unknown variables related to one another. This is a multi-variable problem. When you are dealing with multi-variable problems, the mathematical problems become

more challenging. In this case, fuzzy logic plays an important role in solving this, because we would need to design a similar model, only with a greater number of rules.

Within the geographic division performed by INEGI [4], there is a warm region with extreme summers, including the states of Durango and Nuevo Leon. Durango has reported a negative growth of −1.0 while Nuevo Leon has reported a growth of 3.0. On the other hand, if we were to assess a torrid region as defined by INEGI in its regionalization of the climate seasonality, we would find unequal economies such as that of Mexico City and that of the state of Mexico, with other less developed States such as Guerrero. This analysis confirms what we have been suggesting from the beginning—that we need a comprehensive global analysis, including economic, social and climate variables.

Since the objective of this paper is to show the possible application of fuzzy logic, we have decided to simplify the model and shorten the geographic space to states, taking only political division into consideration. What are the advantages and disadvantages of this? The main advantage relies on a more uniform socioeconomic data (per single state). The main disadvantage is the lack of climate-related information, which decreases the model's precision. This relies, on the other hand, on our main purpose of simplifying the model: remembering that increasing the number of linguistic variables increases the number of rules (exponentially). That is why we have decided to draw this analysis by states and not by climate regions.

## 5. Conclusions

We can reach the conclusion that the method used is pertinent in most of the federate entities when evaluating the EA in each state, especially those with similar bio-climate and/or socioeconomic regions. This method shows that fuzzy logic might be used to measure energy access and to highlight where it is low and deserves special attention.

Nevertheless, to obtain more accurate calculus, we should undertake several possible actions: increase the number of linguistic variables; adjust the values of those linguistic variables; use another survey or even organize our own survey.

We have concluded that the analysis of results by states might be an alternative to the geographic region analysis where the exactitude will depend on the number of variables taken into consideration. Especially when the aim is to implement energy access recovering measures, it is important to precisely define where they will be implemented. It is not the same to define an energy access recovering program for a small city, a municipality or a town as it is for a geographic area in general. That is why it is necessary to include more variables that give a better characterization to the reality of the entities in all-important energy dimensions regarding access.

If we take into consideration all the processing and calculus advantages that fuzzy logic offers us, and combine this with the analysis made, we find out that this approach for evaluating energy access should be taken into consideration by researchers in the field and public policy makers.

The most important result of this paper is that it provides researchers with another tool that has been shown to be useful in the assessment of energy access—fuzzy logic. This technique entails neither high mathematical complexity nor an excessive use of computer time and intensity, while providing a useful model to evaluate and monitor energy access through time.

**Author Contributions:** Conceptualization, D.S.-J. and K.G.C.; methodology, D.S.-J.; software, T.R.-B.; validation, T.R.-B.and K.G.C.; formal analysis, D.S.-J.; investigation, D.S.-J., T.R.-B. and K.G.C.; data curation, T.R.-B.; writing—original draft preparation, D.S.-J.; writing—review and editing, K.G.C.; visualization, T.R.-B.; supervision, K.G.C. All authors have read and agreed to the published version of the manuscript.

**Funding:** This research received no external funding.

**Acknowledgments:** I want to thank the National Science and Technology Council (CONACYT, its Spanish acronym), for the scholarship awarded.

**Conflicts of Interest:** The authors declare no conflict of interest.

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
