# Peer review of "Measurement of Energy Access Using Fuzzy Logic"

_energies, doi:10.3390/en13123266_

Round 1
Reviewer 1 Report
The paper discusses an important topic for the energy literature. More accuracy is needed in the presentation. For example, while the focus is on energy access, the manuscript starts with a phrase mentioning the importance of energy use. In depicting the general context I suggest taking a broader view, looking in general in Latin America comparatively to other parts of the world, and then focusing on the case of Mexico.
"In most indicator sets; the closest indicator is absolute electricity access" - what are the relevant datasets the authors are referring to?
"For instance, the World Economic Forum reports a rate of 100% in 2018 [3]" - where is this rate, everywhere or in Mexico?
A distinction should be introduced between objective indicators and self-reported measures.
More details should be provided about the scientometric analysis (inserting more and better references) and the existing gap. Why do you think that the fuzzy logic was not applied until now?
The empirical application of the paper is clear but the authors should strive to better communicate the results and to integrate them in the existing literature.
.
Author Response
In depicting the general context I suggest taking a broader view, looking in general in Latin America comparatively to other parts of the world, and then focusing on the case of Mexico.
We introduce about energy access in Latin America in Lines 23-38.
"In most indicator sets; the closest indicator is absolute electricity access" - what are the relevant datasets the authors are referring to?
Most of the articles that address the issue of energy access or energy poverty take electricity as one of the main indicators, we clarify this point on line 48.
"For instance, the World Economic Forum reports a rate of 100% in 2018 [3]" - where is this rate, everywhere or in Mexico?
It is in Mexico, it has already been corrected. Line 50.
A distinction should be introduced between objective indicators and self-reported measures.
In this study the database that was taken is a census survey, for which all data is reported by people. We clarify this point on lines 132-135.
More details should be provided about the scientometric analysis (inserting more and better references) and the existing gap. Why do you think that the fuzzy logic was not applied until now?
Within the search that was carried out in web of science, combining both phrases "fuzzy logic" and "energy access" gave a result of zero, so we conclude that there are no articles that talk about this topic, however there are many articles from "energy" where "fuzzy logic" is used. A file with a screenshot is attached showing that the intersection between these phrases is zero. Please see the attachment.
The empirical application of the paper is clear but the authors should strive to better communicate the results and to integrate them in the existing literature.
Thanks for the suggestion, we try to be as clear as possible.

Reviewer 2 Report
Authors : D. Seuret-Jimenez, T. Robles-Bonilla and K. G. Cedano
Title : Measurement of Energy Access Using Fuzzy Logic
Journal : Energies
Paper Number: 761190
An interesting paper and will certainly be beneficial for the Energies readers. The Authors proved the possibility of using fuzzy logic to assess energy access. However, in the case study shown, the choice of factors affecting energy access is confusing. I suggest to justify the choice of variables better. The choice of transport, cooking fuel and electricity expenditure as variables representing of energy access is not convincing. Why is only fuel for cooking included and not, for example, the total energy needed in households? Is heating fuel not used?
Please find below a few minor suggestions:
1. Please remove the dot at the end of the title.
2. Please correct the sentence on lines 11-12.
3. Please correct the sentence on lines 98-99.
4. Please remove extra space in line 114.
5. Please remove extra lines 78, 106, 119, 281.
6. Line 122-124, please discuss why transport is considered as energy access.
7. Figure 2, why is bimonthly and not monthly electricity expenditure presented?
8. Please discuss the correlation (or lack thereof) between electricity and cooking fuel expenditures presented in figures 2 and 3. Maybe electricity is also used for cooking, which reduces cooking fuel.
9. Line 174, it should be “For the present case study, the bands are shown in Figures 4-6.”?
10. In Figures 1 – 3 present states instead of number as in Figure 9?
Author Response
The choice of transport, cooking fuel and electricity expenditure as variables representing of energy access is not convincing. Why is only fuel for cooking included and not, for example, the total energy needed in households? Is heating fuel not used?
We clarify these points on lines 138-142, 144-148 and 183-185. In Mexico, very few people have heating, so it was not considered as an indicator.
According to Sovacool et al [19] “…for both the rural and urban poor, low mobility —regard less of the technology or mode of transport involved— stifles the attainment of better living standards. It reduces the ability to earn income, strains economic resources, and limits access to education and health services and markets…”, is with this consideration that we include transport as a variable related to energy access.
Furthermore, electricity access at this time of world development is crucial, as it provides numerous benefits in addition to other services being closely related, such as entertainment, education, communication, etc. [22]. And, as we know, the massive trend towards electrification will make it more and more relevant to individual and community wellbeing.
It is important to mention that these variables, taken from the literature, are coupled to the variables of the survey, within the survey the expense for household appliances or heating is not included, this is included in the total expenditure of electricity.
- Please remove the dot at the end of the title.Thank you, we remove it.
2. Please correct the sentence on lines 11-12.
Thank you, we correct it. "The regions were determined by the country’s political division (32 states)."
3. Please correct the sentence on lines 98-99.
Thank you, we correct it. Lines 112-113 "Fuzzy logic can obtain results from human language, as opposed to other methods, if the variables and their values are presented as quantifiable data".
4. Please remove extra space in line 114.
Thank you, we correct it.
5. Please remove extra lines 78, 106, 119, 281.
Thank you, we correct it.
6. Line 122-124, please discuss why transport is considered as energy access.
Thank you, we add a few lines, 138-142.
7. Figure 2, why is bimonthly and not monthly electricity expenditure presented?
In Mexico, payment for domestic electric service is made every two months. We add this line in figure caption.
8. Please discuss the correlation (or lack thereof) between electricity and cooking fuel expenditures presented in figures 2 and 3. Maybe electricity is also used for cooking, which reduces cooking fuel.
In this case, the ENCEVI survey does not consider electricity as a cooking fuel. In Mexico, very few households use electric stoves
9. Line 174, it should be “For the present case study, the bands are shown in Figures 4-6.”?
Thank you, we correct it.
10. In Figures 1 – 3 present states instead of number as in Figure 9?
Thank you, we correct it.
This manuscript is a resubmission of an earlier submission. The following is a list of the peer review reports and author responses from that submission.
Round 1
Reviewer 1 Report
Thank you for your research paper. Please find below a couple of comments.
The last phrase of the abstract does not sound academic or professional enough: "This method is recommended because it is easier to use, and it can be expanded or modified since it is a method with not too much computer use". In what sense it is easier, for whom? what about robustness, external validity? etc.
The context of energy use, efficiency and further implications must be expanded in the introduction, with proper references to back up the arguments. In general, the literature must be improved with reliable sources. Suggestions to look up:
Musango, J. K., & Currie, P. K. (2019). Conceptualizing household energy metabolism: a methodological contribution. Energies, 12(21), 4125.
Hoppe, T., Coenen, F. H., & Bekendam, M. T. (2019). Renewable energy cooperatives as a stimulating factor in household energy savings. Energies, 12(7), 1188.Strydom, A.,
Druică, E., Goschin, Z., & Ianole-Călin, R. (2019). Energy Poverty and Life Satisfaction: Structural Mechanisms and Their Implications. Energies, 12(20), 3988.
For the use of fuzzy logic in this research segment:
Zúñiga, K. V., Castilla, I., & Aguilar, R. M. (2014). Using fuzzy logic to model the behavior of residential electrical utility customers. Applied energy, 115, 384-393.
Spandagos, Constantine, and Tze Ling Ng. "Fuzzy model of residential energy decision-making considering behavioral economic concepts." Applied Energy 213 (2018): 611-625.
Ciabattoni, L., Grisostomi, M., Ippoliti, G., & Longhi, S. (2014). Fuzzy logic home energy consumption modeling for residential photovoltaic plant sizing in the new Italian scenario. Energy, 74, 359-367.
Please provide arguments for this particular choice: "We believe that transport, cooking fuel and power expenses were highly representative of the group that should be characterized by its energy access". Especially since in the analysis you argue that "Selected linguistic variables show a similar behavior in all States. This could be an inconvenience for the precision of the method".
Include the computations in section 2.2 in some professional tables, with proper explanations from the Matlab procedures.
Clearly explain the choice and the logic of the Mandani method (with examples and references). Why not Sugeno for example?
Avoid graphs and further analysis in the discussion section. You should include that in your previous section.
Lines 218-223 are fit in the introduction, with proper rephrasing and not in the conclusions.
Reviewer 2 Report
- References are insufficient. Many related papers about fuzzy logic on energy management were missed. For example, the following paper: Mingfu Li, Guan-Yi Li, Hou-Ren Chen, Cheng-Wei Jiang, “QoE-Aware Smart Home Energy Management Considering Renewables and Electric Vehicles,” Energies, vol. 11, no. 9, 2304, Sep. 2018.
- The contributions seem minor and not clearly described in abstract.
- What the membership functions used must be presented in the paper.
- It would be better that different values of High, Medium, and Low are considered and compared, rather than only 0.6, 0.5, and 0.4 are used.
- The descriptions in Section 4 Discussions are hard to read and understand, and must be rewritten.
- The novelties of this work must be addressed more clearly.
- The first paragraph in Section 5 Conclusions should be placed in Section I Introduction.
Round 2
Reviewer 1 Report
Thank you for the improved manuscript. The paper is better positioned but I still believe that the quality of the references used must be increased (with papers published in reputable international journals indexed in SCOPUS and Web of Science). Also, there are still minor spelling mistakes (line 19 importance, line 57 you etc.).
Reviewer 2 Report
I was not satisfied with the authors' responses on my previous comments 1, 4, and 6.
- Almost the references are from local journals, conferences, or websites, and they are not cited in English language. Energies is an international English journal, the authors must cite more international English journals or conferences rather than only local ones in Mexico. The reference format is also not correct. Additionally, the authors' affiliations must also be in English. Notably, references cited in the introduction may not be strongly related the topic of this paper, but they can be included if they can guide readers into related topics this paper has used.
- It would be better that different values of High, Medium, and Low are considered and compared, rather than only 0.6, 0.5, and 0.4 are used.
- The contributions of this paper are minor and this paper maybe has low interest to the readers. How can the results of this paper be applied to other countries?
- There are still some typos in this revision, for example, but not limited,
(1) Abstract: "The regions were determined by the country’s political division (32 states) was the selected method of ...",
(2) "..., this is, ..","..., that is, ..."?
(3) "fussy" or "fuzzy"? They must be consistent through the paper.